# Influence of Microclimate Factors on *Halyomorpha halys* Dehydration

**DOI:** 10.3390/insects12100897

**Published:** 2021-10-02

**Authors:** Francesca Grisafi, Giulia Papa, Mario Barbato, Sergio Tombesi, Ilaria Negri

**Affiliations:** 1Department of Sustainable Crop Production–DIPROVES, Università Cattolica del Sacro Cuore, Via Emilia Parmense 84, 29122 Piacenza, Italy; francesca.grisafi1@unicatt.it (F.G.); giulia.papa@unicatt.it (G.P.); sergio.tombesi@unicatt.it (S.T.); 2Department of Animal Science, Food and Nutrition–DIANA, Università Cattolica del Sacro Cuore, Via Emilia Parmense 84, 29122 Piacenza, Italy

**Keywords:** *Halyomorpha halys*, brown marmorated stink bug, vapour pressure deficit, diapause, nutritional needs, transpiration

## Abstract

**Simple Summary:**

The brown marmorated stink bug *Halyomorpha halys* has become a serious invasive species in Northern America and Europe, where it causes major damage to a wide range of crops. Understanding the ecology and behaviour of this pest is key to identifying the most effective strategies to contain its spread. Here we demonstrate that microclimate conditions affect *H. halys* water loss and that transpiration is influenced by feeding regime and sex. In the overwintering generation, transpiration does not seem influenced by population density and the first nutritional need of individuals exiting diapause is represented by hydration, likely due to water loss during the diapause. Our data suggest that hot and dry climates are not favourable for *H. halys* and may limit its geographical range. Similarly, microclimatic conditions within crops may have a significant impact on the distribution of *H. halys* and insect activity may be affected by crop management practices (e.g., pruning and irrigation).

**Abstract:**

Understanding the interaction between insects and microclimate can be essential in order to plan informed and efficient treatments against agricultural pests. Microclimatic factors such as humidity and temperature can influence the population dynamics of the invasive agricultural pest *Halyomorpha halys*, the brown marmorated stink bug. The aim of this work was to evaluate the level of transpiration of *H. halys* in dry, normal and humid microclimates according to the sex, physiological conditions and developmental stage of individuals. Water loss during diapause and the effect of population density on insects’ transpiration were also assessed, as were the nutritional preferences of adults upon exiting diapause. Our data demonstrate that microclimatic conditions significantly influence the transpiration of this pest species. The effect of sex and feeding status on insects’ water loss is marked, while population density does not influence water loss in diapausing individuals. The first nutritional need of the overwintering generations is represented by hydration, likely due to the water loss during diapause.

## 1. Introduction

The brown marmorated stink bug (*Halyomorpha halys*; Hemiptera, Pentatomidae) is an East Asian invasive species now established across North America and Europe, where it causes significant economic losses to agriculture and ornamental plants due to its polyphagous behaviour [1,2]. To date, most *H. halys* management practices rely on multiple season-long broad-spectrum insecticide treatments [3], which affect both natural enemies and pollinators along with the target pest [4]. Attract-and-kill systems and biocontrol have been suggested as alternative sustainable strategies [4,5,6,7,8,9]; however, such approaches are currently more expensive than the conventional insecticide-based management or pose significant risks of non-target effects, as in the case of the parasitoid *Trissolcus japonicus* [10,11].

Understanding the ecology and behaviour of a pest is key to identifying the most effective strategies to contain its spread. Temperature is known to play a critical role in the biology of *H. halys* [12]. *H. halys* is also a chill-intolerant species, with the minimum and maximum temperature for its development estimated to be 13–14 °C and ~36 °C, respectively [2]. In winter, the species contrasts exposure to cold by aggregating in protected habitats, including human-made structures, and subsequently entering diapause [13,14]. Regions with warm–temperate climates are likely targets of invasion from *H. halys* [15]. The global increase in temperatures due to climate change will further expand the range of *H. halys* and possibly allow bivoltine reproduction in regions where it currently displays only one generation per year [11,15].

Few reports exist on the role of ambient moisture in the life cycle of *H. halys.* Atmospheric and liquid water play a key role in the biology and ecology of terrestrial organisms. Noticeably, small ectotherms have higher risks of desiccation due to their high surface-area-to-volume ratio, low fat storage, and high metabolic rate [16,17]. Although studies on some insect species indicate substantial effects of temperature and relative humidity on transpiration, no data are available on the effects of Vapour Pressure Deficit (VPD) on *H. halys*. Such information can be used to predict the distribution and dynamics of insect populations as demonstrated, for example, in *Alphitobius diaperinus*, the lesser mealworm, a pest of tropical origin that infests grain stores and poultry houses in temperate regions. In this species, the rate of transpiration shows little increase in dry (~0% relative humidity) and warm–hot (20–40 °C) environments, whereas at temperatures >40 °C the insect suffers from high water loss [18]. This confirms that the most suitable microhabitats of *A. diaperinus* are characterised by a low relative humidity and mid-to-high temperatures, such as those found in grain stores and poultry houses [18].

The first response to dehydration in an active insect is to rehydrate by accessing a water source [19]. However, diapausing adults of *H. halys* do not appear to actively rehydrate, though transpiration is not suppressed [20].

Several behaviours can have an impact on transpiration. For instance, the aggregating behaviour of diapausing *H. halys* can act as a water-conserving group strategy, as shown in nymphs of the Pentatomidae *Nezara viridula* [21,22,23,24,25]; however, experimental data focused on the hydration needs of diapausing *H. halys* are currently lacking. Furthermore, transpiration can vary across different stages of the insect cycle [26].

Here, we investigate whether microclimatic conditions can influence the transpiration of *H. halys* and if transpiration is affected by feeding regime, sex and developmental stage (i.e., juvenile or adult stage). Moreover, water loss during diapause and the effect of population density on insects’ transpiration were also assessed, as well as nutritional preferences of adults upon exiting diapause.

This work aimed to expand our knowledge on the ecology of *H. halys* and its reaction to different microclimatic conditions. Prediction of the most suitable microhabitat can aid in the management of this invasive crop pest.

## 2. Materials and Methods

### 2.1. Transpiration during Diapause

We tested the level of transpiration of *H. halys* during diapause and tested its dependency on aggregation size. We sampled four groups counting 10, 24, 50 and 100 individuals, here named G10, G24, G50 and G100, respectively. Sampling of individuals that had recently entered diapause was carried out in late fall 2018, assuring a 1:1 sex ratio within each group, which were kept in beakers (15 cm diameter; Thermo Fisher Scientific Inc. Waltham, MA, USA) covered with insect netting. The beakers were stored in a temperature- and humidity-controlled cabinet (F.lli Galli, Milan, Italy), for 20 days at 8 °C and ~65–70% relative humidity (RH). The net weight of each group of individuals was recorded daily. Weight loss was used as a proxy for water loss (i.e., transpiration) as in overwintering individuals the stored nutrients (lipids, glycogen and sugars) do not decrease during the first months of diapause, but rather at the end (in the spring) [27]. Further, the dry weight of each beaker with covering net was recorded before and after the experiment. To assess the change in weight over time, we fitted Generalized Additive Mixed Models (GAMMs) using the weight measured at each day (Day: from 1 to 20) as a smooth continuous term and the group assignment (Group: G10, G24, G50 and G100) as a categorical variable. We tested models including linear regression, smoothed splines, assigning the group identity as random variable, and day/group interaction. Model selection was performed through Akaike information criterion (AIC) and the model assumptions were verified through graphical assessment of the models’ residuals using the R package ‘gratia’ [28]. GAM was performed as implemented in the ‘mgcv’ R package, which was also used to assess the concurvity and significance of base functions [29,30].

### 2.2. Post Diapause Experiments

Over 800 adults in diapause were collected from the wild between October and December 2017 and stored in cardboard boxes, following Costi and colleagues [31]. The Diapause termination was induced at the end of January 2018 by transferring the boxes to standard rearing conditions, i.e., ~24 °C, ~60% RH and photoperiod 16:8 (L:D) [32].

### 2.3. Double Choice Experiments in Overwintering H. halys Generation

We investigated whether *H. halys* favours rehydration over feeding upon exiting diapause. Upon exiting diapause, 352 specimens of both sexes were transferred to individual petri dishes 12 cm in diameter. Within each petri dish, the insect could choose between free water (a plastic bottle cap filled with water) and a food source (1 g) from among green bean, apple, and sunflower seeds, which are sources of proteins, sugars, and lipids, respectively (CREA ‘Centro alimenti e nutrizione’ https://www.alimentinutrizione.it/sezioni/tabelle-nutrizionali (accessed on 12 June 2020) and Medal and colleagues [33]). We tested 100 individuals for water vs. green bean, 112 individuals for water vs. apple, and 140 individuals for water vs. sunflower seeds. Each petri dish was monitored for one hour to record whether the insect chose to feed or drink. A binomial test was used to evaluate if the choice of water vs. food was significant against the null hypothesis (random choice). Only the individuals which chose either water or food were included in the analysis.

### 2.4. VPD Experiment

We evaluated the level of transpiration of 320 *H. halys* adults across a range of microclimatic conditions ranging from dry (RH < 30%), to normal (RH ranging 30–60%), to humid (RH > 60%). Prior to testing, all the individuals had access to food and water for at least 15 days. To evaluate the effects of starvation on the level of transpiration, 80 individuals were tested after 24 h starvation.

Individual tests were carried out for groups of 10 adult individuals of the same sex. Each group was relocated to a 12 cm petri dish with an insect net lid; we recorded the net weight and moved the petri dishes to the experimental chamber (a 20 cm × 20 cm × 10 cm plastic box).

The RH in the experiment chamber was stabilized within the focal RH range by two air pumps one connected to a silica gel box and the other to a distilled water container, in order to reduce and increase the air humidity, respectively. The experiment chamber was kept at room temperature. Both temperature and RH were recorded every three minutes via data logger (PCE-HT71, PCE Instruments™, Italy). After 40 min, the tested group was removed from the experiment chamber and weighed. No droppings (i.e., marks of excretion) were seen during the experiments. The same experiment was performed on juvenile stages from I to V. We assembled 12 groups of juvenile individuals. To avoid extensive manipulation of the frail juvenile stages we opted to avoid a precise count of the number of individuals per group, and selected approximately hundreds of individuals to balance the small size and the consequent minimal transpiration of each animal [34].

The difference in weight before and after the experiment (Δ*W*) was due to water loss through transpiration, which we expressed as follows:$$Tr=\frac{\Delta W_{G}\cdot 3600s}{2400s\cdot W_{iG}}$$
where *Tr* (mg H_2_O·g^−1^ h^−1^) is the water transpired in an hour on an insect body weight basis, Δ*W_G_* the group weight loss, and *W_i_*_G_ the weight of the group *G* expressed in g.

To account for the relationship between environmental conditions and animal transpiration, mean temperature and mean RH data of each group were expressed in Vapour Pressure Deficit (VPD) [35] as follows:$$VPD=\frac{610.7\cdot 10^{7.5T/\left(237.3+T\right)}}{1000}\left(1-\frac{RH}{100}\right)$$
where *T* is the temperature in °C and *RH* is the relative humidity in %. VPD is expressed in kPa [36]. VPD and *Tr* were subjected to general linear mixed model analysis.

Differences in water loss were assessed by contrasting feeding (fed vs. starving), sex (male vs. female) and age (nymph vs. adult). We modelled the transpiration as a response variable at increasing values of VPD, using the linear mixed model as implemented in the ‘lme4’ R package [37]. We fit two models, the first including adult individuals exclusively and the second, aimed at assessing the effect of age, including both adult and juvenile individuals. Sex and Feed were included as fixed effects in the first model (only adults), and as random effects in the second model (adults and juveniles). The date of measurement was added as a random effect in both models. Model selection was performed through Akaike information criterion (AIC), and the model assumptions were verified through graphical assessment of the models’ residuals using the R package ‘performance’ [28]. Pseudo-R^2^ were computed using the R package ‘jtools’ [38], and confidence intervals on the predictions obtained as ±1.96 times the standard error. Standard errors were computed through 1000 bootstrap replicates of each model using the lme4::bootMer function. A significance threshold of α = 0.05 was used throughout the analyses.

## 3. Results

### 3.1. Transpiration during Diapause

We evaluated the relation between weight loss (as a proxy for transpiration) across the 20 days after exiting diapause in four groups of *H. halys* (G10, G24, G50 and G100). The recorded weights showed a distinct decreasing trend (Figure 1). The initial weight per insect at day 0 was different across groups, with G50 and G24 recording the highest weight and G10 and G100 the lowest (Figure 1). The association between weight and time after exiting the diapause was modelled through GAMM. The lowest AIC was achieved using a log-normal model with Day of measurement as smoothed variable and the group identity as random variable to model the intercept. The Day : Group interaction did not reach significance (*p*-value > 0.05), suggesting non-significant slope differences among groups, and was not included in the final model. We recorded a significant smoothed fit (adjusted-R^2^ = 0.99; *p*-value ≪ 0.01) across the 20 days of the test showing a faster rate of weight loss within the first five days upon entering diapause, followed by an almost linear trend in the subsequent days (Figure 1).

### 3.2. Double Choice Experiments in Overwintering H. halys Generation

In the double choice experiments, out of 352 total individuals 144 chose water (~41%), 23 chose food (~6%), and 185 chose none (~53%). Among the individuals that performed a choice, a binomial test rejected the null hypothesis (*p*-value ≪ 0.01) of a random choice between water and food (Table 1).

### 3.3. VPD Experiment

We assessed the effects of VPD, feeding, sex, and age on the transpiration in *H. halys*. The association between transpiration at changing values of VPD was modelled using a linear mixed model. The lowest AIC was achieved fitting log-normal polynomial regression with VPD (as 3 degrees polynomial term), Sex and Feed as fixed effects, and the Date of measurement as a random effect (total pseudo-R^2^ = 0.64). VPD, Sex and Feed were significant (*p*-value < 0.01). Our model showed a steep increase in the transpiration rate at VPD values 0.5 to ~1.5, and no noticeable increase at higher VPD values, independent of the sex or feeding status of the insects (Figure 2). Independent of sex or feeding status, female and fed showed higher water loss than male and starved individuals (Figure 2b,c).

A second linear mixed model was fitted to determine the difference in transpiration at changing values of VPD between adult and juvenile insects. The lowest AIC was achieved fitting a log-normal polynomial regression with VPD (as 2 degrees polynomial term) and Age as a fixed effect, and Sex, Feed and the Date of measurement as random effects (total pseudo-R^2^ = 0.71). Although the model identified the mean transpiration in juveniles to be higher than in adults across all of the VPD ranges (Figure 3), the hypothesis of different transpiration between adults and juveniles returned a non-significant result (*p*-value > 0.05), likely due to the limited number of measurements we collected for the juvenile insects, as suggested by the large confidence intervals obtained (Figure 3b).

## 4. Discussion

Here, in accordance with observations on other terrestrial species, we found that *H. halys* is sensitive to VPD, with higher values promoting water loss in the species (Figure 2), hence impacting its population dynamics. VPD varies widely according to environmental conditions and plant microclimate as influenced by canopy height and density. Indeed, in maize VPD can range between 0.5 and 2 kPa, while the air VPD is between 1 and 4 kPa [39]. In tree canopy VPD can decrease up to 40% according to the height of the canopy [40]. In our experiment, the maximum VPD tested was 2.5 kPa, which approximately corresponds to an air temperature of 30 °C and RH of 40%, common values during summer in the temperate climates where *H. halys* is spreading.

We found a pronounced effect of sex on the transpiration, as *H. halys* female adults show higher water loss than males, in accordance with previous observations by Ciancio and colleagues [20]. This is in contrast with the behaviour of other species (e.g., dipteran and coleopteran species), where females show increased tolerance to desiccation than males due to a less permeable cuticle, higher body water content, and lower surface-area-to-volume ratio [41,42,43,44,45,46]. The higher water loss displayed by females may be due to their larger size, metabolic rate, total energy content or more permeable cuticle [20,46,47,48].

We recorded a marked effect of physiological conditions (i.e., feeding status) on *H. halys* transpiration, as confirmed by the reduced water loss in starving/dehydrated individuals as compared to fed (Figure 2). Starvation is indeed known to reduce metabolism and transpiration in several species [49,50].

Even if differences were not statistically significant (likely due to the limited sample size of juvenile forms), the values of water loss in *H. halys* juveniles were higher than in adults. Juvenile forms are characterised by a higher surface-area-to-volume ratio and possess a more permeable cuticle than adults [51]. Water loss is also a critical factor affecting the survival of pentatomids nymphs, including *H. halys* [23,26].

Our results suggest that water loss in overwintering *H. halys* first occurs at a faster rate, then decreases according to an almost linear trend [52]. Our observation fits with the known changes in the metabolism rate of *H. halys* through diapause [20]. The beginning of diapause is a specific physiological phase characterized by a transiently higher metabolic activity where the insects are constantly feeding and moving to secure sufficient energy gain for the upcoming long period of metabolic suppression [52]. The following phase is the so called ‘maintenance phase’ of diapause, where an extremely slow metabolism exclusively secures the basic homeostatic functions [52].

We could not record statistically significant differences in the water loss rate across groups of overwintering adults of different sizes (Figure 1). However, it is possible that larger within-group sample sizes are needed to detect the role of aggregation in *H. halys* as already recorded in other gregarious insects, including the pentatomid *N. viridula* [21,22,53].

We found that individuals just exiting diapause seek hydration more than energy resources. According to previous studies, the high mortality rate observed in the overwintering generation is due to a combination of dehydration and metabolic depletion of their energy reserves [54], even if providing water to individuals did not improve overwinter survival [55]. Recently, it has been shown that overwintering *H. halys* adults can exhibit cannibalistic behaviour, likely as an alternative strategy to prevent desiccation, rather than to replenish energy [56].

## 5. Conclusions

We showed that atmospheric VPD affects *H halys* transpiration. Consequently, areas with high VPD (i.e., hot and dry) are not favourable for *H. halys* and may limit its geographical range. Similarly, microclimatic conditions within plant ecosystems, from the canopy to the understory, may have a significant impact on the distribution of *H. halys* individuals. From this perspective, crop management can be oriented to provide an unsuitable environment for the insects, as defined by IPM. The microclimate can be altered by irrigation practices and pruning, which in turn may influence insect pest activity. For instance, sprinkler irrigation washing of foliage can repel the two-spotted spider mite, which favours dry conditions [57]. Similarly, a management strategy incorporating canopy manipulation is effective at reducing infestation of *Drosophila suzukii*, which is heavily influenced by relative humidity [58]. Furthermore, given that the distribution of insect species is known to be limited by areas with high relative values of VPD [59], data on the desiccation tolerance of *H. halys* can improve prediction of the distribution and movement of this important pest under different scenarios of changing climate. Finally, by improving knowledge on specific eco-physiological traits of *H. halys*, our data may promote the development of sustainable control strategies.

## Figures and Tables

**Figure 1 insects-12-00897-f001:**
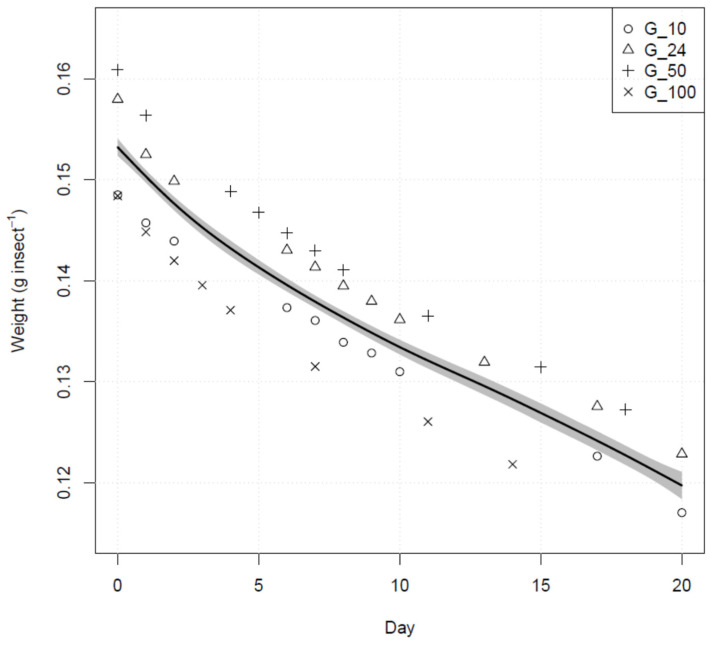
Trend of the mean weight per insect in four groups of *H. halys* with different sample size recorded during the 20 days following diapause. The solid black line represents a smoothed model of the daily weight as inferred through GAMM. The shade grey area represents 95% confidence intervals.

**Figure 2 insects-12-00897-f002:**
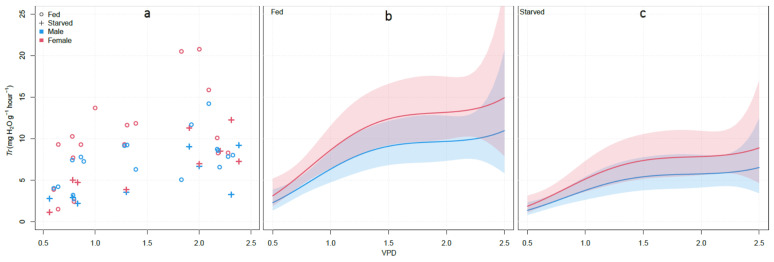
Linear mixed model of the changes in transpiration (*Tr*) of 54 adults at different VPD. (**a**) Scatterplot of the observations recorded and contrasts between males and females under fed (**b**) and starved conditions (**c**). In (**b**,**c**), the solid lines and the shaded areas represent the model prediction and the 95% confidence intervals, respectively.

**Figure 3 insects-12-00897-f003:**
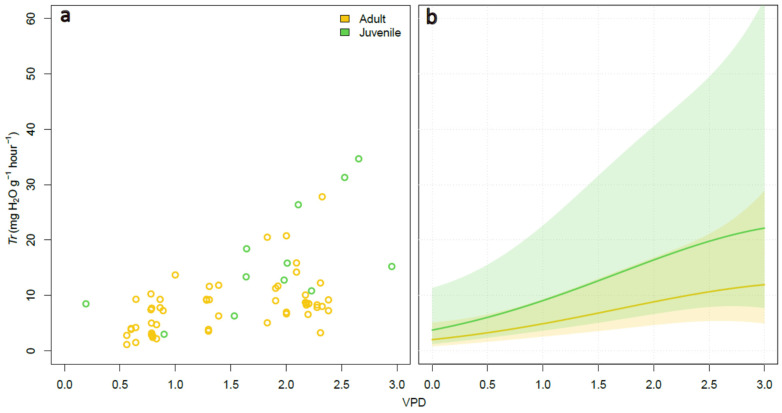
Linear mixed model of the changes in transpiration (*Tr*) of 66 juvenile and adult insects at different VPD. Scatterplot of the measurements (**a**), and model prediction (**b**). The solid lines and shaded areas in (**b**) represent the model prediction and the 95% confidence intervals, respectively.

**Table 1 insects-12-00897-t001:** Choice counts between water vs. food (apple, seed, green bean) and *p*-value of the binomial test.

Choice		*p*-Value	Probability of Success	95% Confidence Interval
Water	144	<2.20 × 10^−16^	0.8622754	0.8005445 0.9106489
Food	23			
Water	44	2.08 × 10^−5^	0.7857143	0.6556046 0.8840778
Apples	12			
Water	64	<2.20 × 10^−16^	0.969697	0.8947834 0.9963089
Seeds	2			
Water	36	6.57 × 10^−5^	0.800000	0.6540417 0.9042427
Green Beans	9			

## Data Availability

The data that support the findings of this study are available from the corresponding author upon request.

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
