# Peer review of "Influence of Microclimate Factors on Halyomorpha halys Dehydration"

_insects, 2021, doi:10.3390/insects12100897_

Round 1

Reviewer 1 Report

The aim of the present study was to evaluate the level of transpiration of an invasive insect, the brown marmorated stink bug Halyomorpha halys, in relation to air humidity, population density, sex, physiological conditions, and developmental stage of an individual. Based on the results of laboratory experiments and observations, the authors concluded that microclimate, sex and feeding status have a significant impact on transpiration and that the water loss represent a serious problem for diapausing adults. These conclusions are not of high fundamental value, because similar studies were conducted and similar conclusions were made on many insect species, but the results of the study can be very important for practice: for the prediction of the further invasion and for the elaboration of best methods for control of this noxious pest.

However, the method used by the authors has a substantial flaw that does not exclude publication but significantly decreases scientific value of the present study. The problem is that in the first, relatively long-term experiment the authors used total weight loss as a proxy for transpiration (water loss) of diapausing adults (lines 175-176). However, decrease in weight of diapausing insects is determined not only by water loss but also by the expenditure of stored nutrients. It is well known that insects entering diapause usually have a large fat body or some other storage of nutrients, whereas during diapause these reserves are dwindling. In particular, a recent study, for some reason missed or ignored by the authors (Skillman, V. P., Wiman, N. G., & Lee, J. C. (2018). Nutrient declines in overwintering Halyomorpha halys populations. Entomologia Experimentalis et Applicata, 166 (9), 778-789), clearly demonstrated the decrease in the content of sugars and lipids in diapausing Halyomorpha halys wintered under quasi-natural conditions. Therefore, to separate the two processes (water loss and loss of nutrients), the first experiment should include separate subsamples used to estimate dry weight, percent water content, content of nutrients, or some other similar parameter. The approximation of water loss by weight loss is OK for the short-term VPD experiment and, moreover, the results of this experiment, in particular, a strong correlation between air humidity and weight loss strongly (but indirectly!) suggested that in this case the weight dynamics was determined by the water transpiration, but the first experiment still remains a problem. Thus, the authors should either conduct additional experiments or consider this problem in detail in the revised manuscript and, if possible, convincingly justify the correctness of their method (e.g. by proving that weight loss in the first experiment was determined only by water loss).

Besides, I have a couple of minor comments.

Lines 2-3: this title does not match the content: in this manuscript almost nothing is said about ‘Mechanisms of adaptation to dehydration...”. I would suggest ‘Factors influencing dehydration....’

Lines 154-155; please, cite the source of this formula.

Author Response

Response to Reviewer 1 Comments

REVIEWER 1

The aim of the present study was to evaluate the level of transpiration of an invasive insect, the brown marmorated stink bug Halyomorpha halys, in relation to air humidity, population density, sex, physiological conditions, and developmental stage of an individual. Based on the results of laboratory experiments and observations, the authors concluded that microclimate, sex and feeding status have a significant impact on transpiration and that the water loss represent a serious problem for diapausing adults. These conclusions are not of high fundamental value, because similar studies were conducted and similar conclusions were made on many insect species, but the results of the study can be very important for practice: for the prediction of the further invasion and for the elaboration of best methods for control of this noxious pest.

However, the method used by the authors has a substantial flaw that does not exclude publication but significantly decreases scientific value of the present study. The problem is that in the first, relatively long-term experiment the authors used total weight loss as a proxy for transpiration (water loss) of diapausing adults (lines 175-176). However, decrease in weight of diapausing insects is determined not only by water loss but also by the expenditure of stored nutrients. It is well known that insects entering diapause usually have a large fat body or some other storage of nutrients, whereas during diapause these reserves are dwindling. In particular, a recent study, for some reason missed or ignored by the authors (Skillman, V. P., Wiman, N. G., & Lee, J. C. (2018). Nutrient declines in overwintering Halyomorpha halys populations. Entomologia Experimentalis et Applicata166 (9), 778-789), clearly demonstrated the decrease in the content of sugars and lipids in diapausing Halyomorpha halys wintered under quasi-natural conditions. Therefore, to separate the two processes (water loss and loss of nutrients), the first experiment should include separate subsamples used to estimate dry weight, percent water content, content of nutrients, or some other similar parameter. The approximation of water loss by weight loss is OK for the short-term VPD experiment and, moreover, the results of this experiment, in particular, a strong correlation between air humidity and weight loss strongly (but indirectly!) suggested that in this case the weight dynamics was determined by the water transpiration, but the first experiment still remains a problem. Thus, the authors should either conduct additional experiments or consider this problem in detail in the revised manuscript and, if possible, convincingly justify the correctness of their method (e.g. by proving that weight loss in the first experiment was determined only by water loss).

Our response.

In the article by Skillman and colleagues (2018), it has been demonstrated that in overwintering adults the weight, glycogen content and sugar levels start to decline at the end of the diapausing period i.e, from March-April (spring); on the contrary, during the first months of diapause (i.e., from October to January), the levels do not vary significantly (Skilman et al., 2018). Our experiment has been carried out in late fall, and specifically in late November, thus no decline in nutrient storage should be expected. We clarified this in the text.  Interestingly, from November to December, Skillman and colleagues even found a significant increase in the lipid content of overwintering individuals while the differences in weight, glycogen, and sugar levels were not significant. This would even mean that the water loss of our populations might be more than expected, but further work is of course needed to confirm this.

Besides, I have a couple of minor comments.

Lines 2-3: this title does not match the content: in this manuscript almost nothing is said about ‘Mechanisms of adaptation to dehydration...”. I would suggest ‘Factors influencing dehydration....’

Our response.

We thank the reviewer and following his/her suggestion we changed the title into “Influence of microclimate factors on Halyomorpha halys dehydration”

Lines 154-155; please, cite the source of this formula.

Our response.

A reference has now been added (Castellví et al., 1996).

Author Response

Response to Reviewer 2 Comments

REVIEWER 2

Major comments: The paper is what was prevalent in the older literature (before 1980), although the terminology was not exact in this paper.

Diffusion was used to indicate insect movements but diffusion is the mode of water loss across cuticle.

Our response.

We thank the reviewer and corrected the mistake: “diffusion” was changed with “spread”.

The clarity of the equations used in the methods was mixed as one used TWi, but that term cannot be correct as it is mass loss per gram of insect that is calculated. Also, more clarity of what each term means in the second formula is needed as it is not clear whether the calculation of VPD is correct.

Our response.

The section was revised, the acronyms were changed and their description was improved.

Figure 1 is what usually observed when water that adheres to the cuticle is not removed to begin the experiment (usually 1 day in the conditions prior to starting the experiment).

Our response.

The experiment of Figure 1 lasted 20 days. Thus, the weight loss cannot be due to the evaporation of water adhering to the cuticle.

What does it mean “The whole dataset was cleaned from data attributable to instrumental errors” if all you were doing was weighing the insects (lines 158-60)?

Our response.

We removed the sentence since it was referred to instrumental errors that occurred one time due to a machine failure.

Finally, “evapotranspiration“ is a specific term that you did not measure, only weight loss was measured.

Our response.

we agree with the reviewer and changed the term “evapotranspiration” with “transpiration” throughout the text.

Insects that were not in diapause could have excreted as well. You did not indicate if that might have been the difference between “Fed” and “Starved” insects. 

Our response.

We didn’t see any droppings left by individuals in the cages, so we may hypothesize that no excretion occurred during the experiments. We clarified this in the text.

Potentially reading E.B. Edney’s book “Water Balance in Land Arthropods” from Springer (1977) would help clarify the older information.

Our response.

We thank the reviewer for this suggestion. We believe that old literature, and specifically studies regarding physiology and behaviour, represents a fundamental starting point to improve our knowledge especially on alien pests.

Minor comments

Line 56. Entering diapause

Our response.

We thank the reviewer and corrected the mistake

Line 57 Change “target” to plural

Our response.

We thank the reviewer and corrected the mistake

Line 67 Change “population” to plural

Our response.

We thank the reviewer and corrected the mistake

Line 68-72 Unclear example of what the water loss of lesser meal worm actually shows. Need to make this clearer of how this shows VPD is important rather than just temperature effect.

Our response.

We rewrote the sentence

Line 82 “stage” should be plural

Our response.

We thank the reviewer and corrected the mistake

Line  89-90 Unclear what sentence means

Our response.

We rewrote the sentence

Line 102 Should “model” be added after GAMs

Our response.

We added “model” after GAMs.

Line 111 Need to insert “were collected”

Our response.

We corrected the mistake.

Line 122-23. Need to indicate water content of various foods used in choice experiment

Our response.

We believe that this information may be useful only in the context of double choice experiments between different food resources but not in experiments between food and water.

Line 129-30 Need to change “microclimates” to adjective or remove “conditions”

Our response.

We thank the reviewer and corrected the mistake.

Line 142 Change “weighted” to “weighed”

Our response.

We thank the reviewer and corrected the mistake.

Line 232-3. Only “water loss across the cuticle” is relevant to this work, as well as potential respiratory water loss

Our response.

we agree with the reviewer and changed evapotranspiration with transpiration.

Line 243-45. Not all insects evaporative cool, so unless the authors have shown that this insect does, the references are not supportive of the statement.

Our response.

We agree with the reviewer and deleted the sentence and the relative bibliography.

Final conclusions may be correct, but the host plant would ensure that conditions are not dry relative to petri dish, so the paper would be improved if VPD within a plant was measured.

Our response.

We added a paragraph providing information about the range of VPD that can be measured in the environment and in plant canopies, with references (Bachofen et al., 2020; Cavero et al., 2009).

VPD in locations where the insects form groups prior to diapause would also help in understanding where on the graphs those conditions occur

Our response.

We agree on the importance to know exactly which kind of habitats (crops, wild plants, …) are preferred by H. halys. This is indeed the aim of a specific study that is ongoing on the spatial distribution of H. halys populations in different agroecosystems according to different VPD values.

Reviewer 3 Report

The experiment was done well and the data were treated correctly. Only the formulations in the introduction and some other parts show that the authors are not familiar with this theme of reserch, some sentences are incorrect of strange.

Round 2

Reviewer 1 Report

Although I still believe that data on dry weight would substantially increase the quality of the present study, I am satisfied with the explanations provided by the authors and with the corresponding changes to the manuscript. Thus, the paper can be published.

Author Response

We thank the reviewer for his/her comment. 

Reviewer 2 Report

see attached form

Author Response

We thank the reviewer for his/her suggestions.

Our responses are listed below:

  • We corrected typos and errors on lines 86, 110, 140, 168, 179, 230, 260, and 290.
  • We rewrote the sentence on lines 144-145 to improve the clarity of materials and methods.
  • The size of the datalogger (PCE-HT71, PCE, Italy) was 106 x 24 x 24 mm.
  • The number 610,7 in the formula is a constant (Tetens, O. 1930. Über einige meteorologische Begriffe. Z. Geophys 6: 297-309; Monteith, J.L., and Unsworth, M.H. 2008. Principles of Environmental Physics. Third Ed. AP, Amsterdam; Castellví, F.; Perez, P.J.; Villar, J.M.; Rosell, J.I. Analysis of methods for estimating vapor pressure deficits and relative humidity. Agricultural and Forest Meteorology 1996, 82, 29–45).
  • We agree with the reviewer that insects that did choose water needed to rehydrate while we cannot say anything about insects that did not choose food nor water.
  • We agree with the reviewer that since females are bigger than males, they may be more subjected to water loss. Regarding the cuticle, no data are available on the permeability of the cuticle of females and males. However, since that both size and a more permeable cuticle may actually affect transpiration, we added this in the text.